# A Focus Group Study of Perceptions of Genetic Risk Disclosure in Members of the Public in Sweden: “I’ll Phone the Five Closest Ones, but What Happens to the Other Ten?”

**DOI:** 10.3390/jpm11111191

**Published:** 2021-11-12

**Authors:** Carolina Hawranek, Senada Hajdarevic, Anna Rosén

**Affiliations:** 1Department of Radiation Sciences, Oncology, Umeå University, 901 87 Umeå, Sweden; anna.rosen@umu.se; 2Department of Nursing, Umeå University, 901 87 Umeå, Sweden; senada.hajdarevic@umu.se; 3Department of Public Health and Clinical Medicine, Family Medicine, Umeå University, 901 87 Umeå, Sweden

**Keywords:** cascade testing, family communication, hereditary cancer risk, cancer prevention, genetic counselling, personalized medicine, focus groups, qualitative methodology

## Abstract

This study explores perceptions and preferences on receiving genetic risk information about hereditary cancer risk in members of the Swedish public. We conducted qualitative content analysis of five focus group discussions with participants (*n* = 18) aged between 24 and 71 years, recruited from various social contexts. Two prominent phenomena surfaced around the interplay between the three stakeholders involved in risk disclosure: the individual, healthcare, and the relative at risk. First, there is a genuine will to share risk information that can benefit others, even if this is difficult and causes discomfort. Second, when the duty to inform becomes overwhelming, compromises are made, such as limiting one’s own responsibility of disclosure or projecting the main responsibility onto another party. In conclusion, our results reveal a discrepancy between public expectations and the actual services offered by clinical genetics. These expectations paired with desire for a more personalized process and shared decision-making highlight a missing link in today’s risk communication and suggest a need for developed clinical routines with stronger healthcare–patient collaboration. Future research needs to investigate the views of genetic professionals on how to address these expectations to co-create a transparent risk disclosure process which can realize the full potential of personalized prevention.

## 1. Introduction

The understanding of hereditary cancer susceptibility genes has opened new opportunities for personalized cancer prevention and control through tailored screening programs, frequent monitoring and risk reducing surgeries [1,2,3]. However, the implementation of genomics-driven cancer prevention has proven slower than desired [4,5], suggesting a need to align practice with patient needs and clinical contexts. International clinical guidelines warrant pre-symptomatic genetic testing in high-risk families for tier-1 conditions [6] such as hereditary breast and ovarian cancer (HBOC) and Lynch syndrome (LS). The cost-effectiveness of such targeted screening for hereditary cancer is well established, but outcomes are dependent on the number of relatives reached for each tested index case—the first person in the family in whom a genetic risk is confirmed clinically [7,8].

In Sweden and most other countries the standard practice for informing at-risk relatives is the family-mediated approach, where the index case is relied upon to pass on risk information to relatives [9,10]. This approach is not very effective, as research shows only around one-third to a half of those at-risk are successfully informed about a potential genetic risk [11]. Factors impeding family-mediated disclosure include family conflicts, worry about upsetting others, selective informing, misunderstandings and forgetfulness [12,13,14].

Evidence on best practice for risk disclosure is limited [6,15], and lack of consensus in practice is exemplified by how pre-test consent forms vary widely in content [16]. Interventions to support family-mediated disclosure have so far failed to convincingly increase uptake of genetic testing in relatives [17]. Efforts evaluated include motivational interviewing [18], dedicated telephone follow-up [19], offers of genetic testing [20] and online cascade-testing interventions [21]. Although international guidelines have moved towards allowing direct contact in select circumstances [22], healthcare professionals have been reluctant to adopt new proactive practices as legal and ethical regulations remain uncertain and are not easily resolved [23,24].

A growing body of research indicates that a large majority of people in general want to receive actionable genetic risk information, and the “right to know” is something most people feel strongly about [25,26,27,28]. Public attitudes towards cancer genetic testing are also very positive [29,30], and high acceptance for direct contact approaches where healthcare professionals (HCPs) approach eligible relatives has been reported in Denmark, Sweden, France and the US [29,30,31,32]. Factors associated with the desire to be informed of a genetic risk include the emotional relief when reducing uncertainty and the clinical usefulness of individual test results [25,28,33,34]. Although the family-mediated approach has been described as the preferred choice by patients and relatives, a more active involvement by healthcare professionals in the disclosure process is solicited [27,35].

Attitudes towards hereditary risks have been mainly explored in cancer-affected populations or mutation-carriers [17,23,35,36] or studied in the context of the return of genetic data from research [26], but knowledge of how healthy, uninformed and previously unaffected people in general perceive sudden news of hereditary cancer risk is still limited.

In this study we therefore explored perceptions and preferences of the disclosure of hereditary cancer risk among members of the public who could be potential at-risk relatives. We particularly focused on how unaffected individuals view current clinical practice, how they would like to be approached or contacted about a potential hereditary cancer risk and how these insights could lead to meaningful improvement of current cancer genetic care.

## 2. Materials and Methods

We used a qualitative approach to capture the variation in public perceptions of receiving and discussing hereditary cancer risks. In this study the terms “disclosure” and “communication” are used interchangeably and refer to receiving or sharing information about an increased clinically confirmed genetic risk. All discussions in this study related to scenarios with treatable cancers, such as breast or colorectal cancer, that may or may not be inherited by subsequent generations. The term “index case” refers to the first individual in a family in whom a known pathogenic variant is confirmed by genetic testing. During the vignettes we tried to describe these phenomena in layman’s terms and offered time for questions in order to verify the understanding in our participants (Appendix A). 

We used focus group discussions (FGDs) which are well suited for exploring how attitudes and opinions are formed and for researching sensitive topics, as the setting among peers can create a safe and encouraging environment for participants [37]. The focus group format allows subjects to express themselves in their own vocabulary and elaborate on each other’s statements through interactive dialogue [38]. The method minimizes the influence of the facilitator and is advisable in settings where there is a gap between professionals and target audiences or power differential between respondents and decision-makers [39].

### 2.1. Participant Selection

We targeted members of the public with a variety in age and educational backgrounds selected to be naïve to the situation of receiving genetic risk information to be able to share instinctive perceptions. Participants were recruited through snowball sampling from different social contexts: a local public day-care center, a canine agility training club, a facility technical support office, a public authority workplace and a beach volleyball association. Individuals identified by the research team were approached with a request to ask peers in their community if they could participate in a group interview. Short, written information with contact details to the research team was distributed if needed, and a time and date was scheduled to suit the participants who volunteered.

### 2.2. Data Collection

Data were collected in a total of five sessions. The first four were organized in May through August of 2017, and one additional session was added in October 2019 to confirm saturation and capture potential shifts in public opinion over time. We used a small group size of 3–4 participants per session to ensure a comfortable atmosphere for potentially sensitive personal discussions and provide each participant with ample time to reflect and elaborate on their thoughts [40]. Four groups were mixed gender, and one had all male participants. Interviews were held in a neutral meeting room in office-like administrative premises at the University Hospital of Northern Sweden. The duration of the discussions ranged between 79 and 94 min (median = 85 min).

The focus group interviews were initiated with a brief presentation of participants and facilitators, followed by a very short introduction to the clinical context under study and how genetic testing can identify hereditary cancer risks. An interview guide with 13 questions was developed based on literature review and input from clinical professionals and research colleagues (see Appendix A). The discussions started with very broad open questions such as “Would you want to find out about a hereditary cancer risk running in your family?” The group was invited to reflect on their views regarding if and how they would like to receive such information. The facilitator used probing questions to elaborate and clarify statements if needed and finally presented a few hypothetical scenario vignettes to prompt discussion about situations encountered in genetic counselling [41]. Towards the end of the sessions, the facilitator reiterated a summary of the discussed topics to confirm the understanding of her discussion with the participants. 

Discussions were held in Swedish and facilitated by two authors per session. The main facilitator (S.H.) with experience in qualitative methodology led all sessions assisted by a second colleague, specializing in genetics or biomedicine, who collected consent forms and managed the audio equipment. Interviews were recorded, transcribed verbatim, cross-validated with audio and anonymized. No incentives, financial or otherwise, were used in this study.

### 2.3. Data Analysis 

The dataset was analyzed using qualitative content analysis [42]. All authors read the entire transcripts several times to establish a sense of the whole. The main coder (C.H.) identified meaning units related to the research question and condensed and applied descriptive codes. All authors met for recurrent analytical discussions to establish consensus on codes. All codes were grouped into clusters with similar meaning, distinct from other clusters, to form sub-categories (Table 1). Related sub-categories were then grouped and interpreted to form main categories. Consensus meetings were held to establish a coherent level of interpretation, decide category phrasings, and validate preliminary findings in underlying data. Coding, decontextualization and recontextualization up to the sub-category level was performed in the software OpenCode 4.03, a freeware developed for qualitative analysis (17).

## 3. Results

The final sample consisted of eight men and 10 women aged between 24–71 years (Table 2). The participants did not have any previous experiences of genetic counselling. Educational background varied from primary school to completed academic degree. Occupations ranged from being employed in preschool education, banking, academic research, photography, retail, nursing, administration, facility management or being a full-time student or on parental leave.

Overall, participants described the topic of hereditary risk disclosure as difficult and complex. In parallel, they recognized the importance of such information and how it could positively affect people’s lives, but also cause harm in the form of worry and distress.

### 3.1. Overall Theme: A Maze of Challenges in a Haze of Silent Expectations

The overall theme of meaning offers an interpretation of the full dataset. The maze symbolizes the difficulties of navigating among one’s own values, perceived demands and worries about choosing the right path. A maze full of crossroads also illustrates the multitude of strategies and decisions respondents considered while not being able to clearly discern where their chosen path of action might end. 

The haze illustrates the ever-present uncertainties characterizing almost every topic in the discussions: challenges of understanding genetic risk levels, not knowing how relatives would react, being unsure what clinical support is available and questioning one’s own convictions in certain situations. The silent expectations refer to the unspoken assumptions and thoughts about how other stakeholders ought to act and perceived expectations anticipated by our respondents. In the haze it is difficult to distinguish the moral boundaries, blurring the lines between duties and conflicting rights of the different stakeholders involved (index case, relatives, and healthcare). These hazy moral boundaries add to the ambivalence of wanting to know or not and wanting to do the right thing but being held back by worry and fear of doing it wrong.

Two underlying descriptive themes further delineate the findings: “face an important but difficult challenge” and “expect healthcare to lead but also support disclosure”. The overall and descriptive themes give direction and nuance to the latent content in the text and describe the red thread connecting the associated categories (Table 3).

### 3.2. Theme: Face an Important but Difficult Challenge

The first theme covers three categories describing perceptions of likely emotional challenges, importance of inter-personal relationships and the recognition that disclosure of hereditary risk is difficult and requires professional skills to be managed well.

#### 3.2.1. Struggling with Unpleasant Feelings and Consequences

This category reflects both the immediate envisioned reactions when being told of a genetic risk of disease and the subsequent situation of having to “pass on the bad news”. Participants intuitively switched between the perspectives of receiving risk information and disclosing information to a relative. They recognized how each situation could spark anxiety and worry on behalf of loved ones and unease in being the one expected to discuss sensitive health issues with relatives.

The perceived difficulties when sharing risk information within families included the fact that the topic may be sensitive and cause worry, that one may be worried about relatives’ reactions or have bad timing when contacting someone. To reduce the risk of harming close relatives and protecting them from unnecessary worry, a number of strategies were discussed by participants. Examples included to wait for the return of a genetic test results before telling family members or to meet in person during holidays to be able to discuss and handle questions which may arise.

*“I’m thinking it would be a terrible responsibility for the one who is 23 and in charge of informing the entire family”*…(*K1, FGD1*)

Participants described a fear of prompting a lifelong worry of disease in healthy people and acknowledged the fact that some people may prefer to stay oblivious to genetic risks. In parallel with the envisioned difficulties and concerns about disclosure, participants also described a strong conviction that warning individuals at risk of hereditary disease is “the right thing to do”. Access to preventive care was described as the main motivator to go through with disclosure, despite the negative feelings it may cause in oneself.


*“I think it is obvious. One cannot walk around and keep it inside because they need to get their check-ups too—they must get checked.”*
(*M2, FGD4*)

#### 3.2.2. Allowing Type of Relationship to Govern Preferences

Most participants said they would feel responsible for informing relatives if given an informative result from a genetic test. Some described the wish to warn family about the upcoming news but preferred the risk disclosure itself be performed by an HCP. Others said they would definitely inform their family themselves, or would want to hear the news from relatives, but would then want swift access to a genetic professional who could answer questions promptly and accurately as soon as they arise.

The data show a clear distinction between preferences on how to communicate with close relatives and more distant relatives. When considering disclosure to children, siblings and parents, the spread of information was often considered urgent, especially when children could be at risk. More distant relatives such as cousins, aunts and uncles were seen as a challenge to contact, and several reasons were mentioned as to why not reaching out to them would be reasonable or excusable. The “closeness” was not always defined purely as genetic. The “distance” in relationships could also be geographical or social: having fallen out with a sibling or lost contact with someone who has moved abroad.

*“Closest ones I could also call, that’s fine, to talk to by myself, or when we get together. I don’t know, but it may feel really difficult to call…well like auntie whom I haven’t seen or talked to for 20 years”*…(*K1, FGD3*)

Some respondents made a point of the fact that a close relationship could in some cases also hinder disclosure. For example, a young daughter not wanting to talk about private female health issues with her father or familial norms affecting which subjects are considered taboo or “something we would never talk about”. Thus, the quality of inter-personal relationships determined the preferences and imagined actions described in each single case.

In contrast, there was no difference in preferences on which mode of disclosure was best in regard to one’s role or perspective, i.e., being either the relative receiving risk information or being the index case expected to spread risk information. Several participants changed their imagined role mid-argument, just to clarify their stance or preference from both sides.


*“…say you are my sister, and I call you to tell this, but then you ask me follow-up questions and I’m not educated to answer or calm her. Or…to explain what will come of this, how big is the chance…well, that’s it really, if you would tell me, then I would have follow-up questions and then I think my sister would be very fazed, and she’s not educated to handle if I become shocked…”*
(*K1, FGD2*)

#### 3.2.3. Feeling That Risk Disclosure Requires Skill and Experience

No matter what stance participants took on best mode for risk disclosure there was a consensus overall that risk disclosure is a complex and difficult issue to discuss. On the one hand, family-mediated disclosure was considered the best option for some people, in some situations and for certain relatives. Being able to judge good timing and knowing the person’s mental state or recent family affairs were discussed as factors which could contribute to a good disclosure experience. On the other hand, the uncomfortable situation of being the “messenger of bad news” but lacking the professional skills to handle potential distress or shock were debated in detail. Concerns about putting the individual informing in a stressful situation when confronting their relatives were repeatedly addressed.

The relative receiving unexpected risk information was assumed to need fast access to a healthcare professional, as the layperson would not have the skill or expertise to answer questions. Pre-scheduled appointments, hotlines to genetic counsellors or in-person disclosure meetings with healthcare staff present were described as ways to minimize the time of uncertainty until one could consult someone at the clinic. The genetic professional answering these initial calls was expected to have good insights into the individual case and be able to answer detailed questions right away.

Most participants considered risk information important enough to set their own discomfort aside. However, hesitant statements about what to say and how to say it were common as well as statements questioning the suitability of relying on laymen to communicate complex genetic risk information. Respondents described feeling unfit to inform about risk by themselves, soliciting active support or presence of a healthcare professional when speaking with relatives. Feelings of responsibility to “get it right” and not upset the relative were voiced as well as concerns about negative consequences if one could not handle the reaction or emotional response elicited by the risk information.


*“No, but I think such information is best conveyed by knowledgeable staff; there is a risk that something is distorted or misinterpreted if it should go through someone…”*
(*M1, FGD 5*)

### 3.3. Theme: Expect Healthcare to Lead but Also Support Disclosure 

The second theme includes data revolving around preferences on how hereditary risk information should be managed and ideas about how healthcare institutions should support and coordinate the process of disclosure. 

#### 3.3.1. Depending on Healthcare to Take Main Responsibility

This category outlines the participant’s ideas about their own contributions versus healthcare’s duties to safeguard that risk information reaches all those affected. Participants voiced a desire to share the responsibility to inform between the index case and healthcare staff for instance by listing relatives together and then deciding with the counsellor who will contact each relative. Participants made it clear that they would want to be involved in the decisions on whom to inform and how, and some wanted the option of completely handing over the task of risk disclosure to healthcare if one did not feel comfortable.


*“It (family-mediated disclosure) is a good complement, of course one should talk about it, but at the same time information should be disclosed by healthcare too…because there one can only encourage the person to talk about it, while if healthcare manages disclosure, it will reach them…”*
(*K2, FGD 2*)

In the situation where participants were asked to imagine being the one disclosing risk information the need for expert advice and mentoring was described in different envisioned approaches. Some wanted to have an HCP by their side when informing, have the doctor host a family gathering or even make house-calls to inform. At the same time, many recognized these ideas as unrealistic due to limited resources. Some made it clear that informing family on their own was not an option—especially not by telephone. They expressed a need for continuous cooperation with healthcare staff to do their part, for instance by forwarding a prepared information letter drafted by healthcare.


*“I feel that I would not want to be the one who tells them. I would prefer that it came from you to them. I mean, I wouldn’t want to challenge my siblings to go for an appointment…but you could send the same letter to them.”*
(*K1, FGD2*)

Although for some disclosure was considered manageable and straightforward, this view often changed as discussion moved to more distant relatives such as cousins, where barriers to disclosure were anticipated and discussed. These discussions resulted in the assumption that if family communication failed at least healthcare would step in and offer some form of backup plan to safeguard that the uninformed are indeed reached.

#### 3.3.2. Surprise and Frustration about Current Practice 

In each group the facilitator explained the current standard practice of family-mediated disclosure. Reactions ranged from concerns about individual consequences if information fails to outright questioning of the legal and ethical position of the current praxis. Participants were also surprised when learning that in clinical practice there is currently no systematic follow-up of which at-risk relatives have been reached. Respondents expressed that healthcare should take the main responsibility to make sure at-risk relatives were reached by some sort of basic information, so that they would be able to make their own choices about genetic testing.


*“So, I’ve gotten this result now, and I’m supposed to sit down and phone around…and I might not know the relatives that well, so maybe I’ll phone the five closest ones, but what happens to the other 10?”*
(*K1, FGD3*)

The “right to know” was brought up, and some participants argued that they have a right to what they considered was “their own genetic data” even if the data are derived from testing of another family member. In several groups participants suggest that genetic risk disclosure should be covered by similar legislation to that governing risk disclosure when people have been subjected to certain infectious diseases—where one is obliged by law to contact at-risk individuals. 

Towards the end of each interview session the groups were presented with a blocker scenario, meaning a person who refrained from contacting at-risk relatives, leaving family members unaware of a potential risk of developing cancer. Participants expressed that they would probably be angry and frustrated if in the shoes of such an uninformed relative and then become ill without having had the opportunity to access preventive measures. Some groups arrived at the conclusion that in a blocker situation the duty to ensure that accurate risk information reaches concerned relatives lies with healthcare.


*“Yeah, but sure I would think…I think I’d be really pissed off actually if I found out I could have known…and was not given the chance…”*
(*K1, FGD 1*)

#### 3.3.3. Wanting Personalized Information to Empower Informed Health Choices

A clear need for understandable and tailored information was justified by the desire to be given the opportunity to make an informed choice or give that same opportunity to family members who may be at risk. Motivations behind receiving risk information was generally framed around possibilities for prevention, especially in children and close relatives. Peace of mind was another motivator, and the notion that knowing your genetic status would be positive in itself. 

Those who preferred being informed by a family member still had a number of envisioned services healthcare could offer in parallel to the relative informing them. They wanted the initial information paired with swift access to a genetic professional for further questions and clear instructions on how to get counselling and start the testing process and discussed several forms of supportive written information. Several groups briefly discussed the desire to be informed as being connected to “being able to do something about it”. For some, clinical utility was a prerequisite for the desire to be informed, while others expressed a general wish to know of hereditary risks. In parallel several groups acknowledged that some prefer a carefree life without having to be concerned and instead would “stick their head in the sand”. The wish to be able to make an informed choice was described as critical, being more important than the mode of disclosure for instance.


*“If I receive it from a good physician, I don’t think I would feel offended by the family member who had not told me because it’s a tough thing to tell. Yeah, I don’t think it would make any difference, or it would, but it would not change my intention to pursue testing myself…”*
(*K1, FGD5*)

When the discussion concentrated around practical aspects of being informed, participants voiced the need for clear, concise and highly personalized risk information. Visual aids, colors and pie-chart diagrams based on clinical data on one’s own family and available risk estimates for the individual were described as something that would improve understanding.


*“…that’s why I think a small folder like this is the way it looks, so that one gets like a visualized family tree…almost with red lines…”*
(*M1, FGD3*)

Participants also highlighted the need for just enough content, preferably presented in a stepwise manner as to allow for gradual understanding of the situation. In addition, participants commented on the importance of the tone of voice in letters, pointing out that the focus should be placed on positive content to induce hope and preventive action instead of scaring people and causing worry or denial.

## 4. Discussion

We outline instinctive perceptions in people naïve to the situation of receiving and sharing genetic risk information. Our results illustrate the complexity of disclosure dilemmas and highlight preferences on risk disclosure from the perspective of unaffected members of the public. Despite our attempt to focus on the role of receiving genetic risk information, our participants automatically took on the perspective of someone disclosing a genetic risk to family. They tended to discuss the two roles interchangeably to motivate their preferences and did not see the opposing roles as a factor influencing these, and neither did they approve of a dualistic view on either using direct or indirect contact to reach relatives. Family communication or healthcare-assisted risk disclosure was not considered contradictory but rather complementary practices that need to be coordinated.

### 4.1. A Complex Topic Eliciting Worry and Concern for Others

Our data confirm earlier research showing that the topic of genetic risk disclosure can elicit a variety of unpleasant feelings, such as being worried about upsetting others, fear of a relative’s reactions and feelings of guilt and anxiety [43]. In a Dutch study on cancer patients and unsolicited genetic findings, two of four main themes addressed the cognitive and emotional complexity of this information [44]. In an interview study on family communication of CRC risk in Canada, one in four index patients had not told a single relative due to the anticipated negative reactions in relatives [45]. 

Another aspect voiced in our data was the uneasy feeling of responsibility for someone else’s health. This has been described earlier in patients, in terms of “a burden” or “an obligation” [43,46]. In the scenario of being informed of a hereditary risk, again our respondents were quick to consider the implication for family members. The strength, closeness and the quality of relationships with a specific relative were often the main considerations when deciding on their preference of either sharing risk information or not. Timing, current health status etc. were also considered but described as having less importance. 

As a rule, our interviewees acknowledged the importance and urgency of sharing health-related genetic information among relatives who might benefit from such knowledge. However, they also identified several emotional and logistic challenges, such as geographical distances, lack of social contact and tense relations which have previously been described in patients and genetic counselees [31,36,43,47,48]. One way to understand the swell of negative emotions in these situations is that they are caused by conflicting duties. On one side respondents want to alert family members of a danger, and on the other they want to protect them from hurt in the form of anxiety and fear [15]. 

Our respondents indicated that a way to dissolve the negative feelings is to act on them as fast as possible, for instance, calling an HCP right away or getting tested as fast as possible. This reaction could reflect a type of coping strategy where action is a way to gain a sense of control [44,49]. A Swedish study on intentions to take a genetic test describes two alternative coping strategies: the “monitoring” personality who seeks out and acts on risk information and the “blunting” personality who avoids or denies risk [50]. Indeed, both behaviors were envisioned and described by our participants.

### 4.2. Sharing the Responsibility by Collaboration with HCPs

Our data also corroborate previous findings that people indeed let quality and type of relationship govern attitudes and disclosure preferences [47,51]. A systematic review suggests index cases make clear distinctions about their moral obligations based on the kind of relationship they feel they have with different at-risk relatives [46]. Limiting one’s own perceived duty to inform only selected individuals could be interpreted as a compromise to escape the discomfort of conflicting moral duties, i.e., the duty to warn and the duty to protect from harm. By defining one’s own “*genetic responsibility*” based on family ties our respondents could be said to refer to the affective/relational approach to moral responsibility as opposed to the rational/principle-oriented approach discussed when placing part of the responsibility on healthcare [46].

It is important to note that the preference to inform only selected relatives did not imply that our respondents wanted to relinquish the responsibility of risk disclosure altogether. The question was not about selecting one or the other mode of disclosure as the optimal approach but rather using a mix of both family-mediated and healthcare-mediated disclosure efforts, preferably with healthcare taking the main responsibility. Wishes for more active collaboration with HCPs has been reported in studies of Canadian cancer patients [28], US patients with hereditary cancer [31] and in patients, relatives and the public in the Netherlands [27]. In our data several constellations and approaches were envisioned. Among those, a recurrent idea was to give a “heads-up” to the relative and then allow the HCP to distribute a formal letter, or to agree beforehand on who would contact which relatives.

The wish to collaborate around disclosure was in part also motivated by a sense of inadequacy according to our respondents. Lack of skills to discuss genetic issues or handle potential emotional reactions, in oneself or in other family members, was seen as an obvious obstacle to successful risk communication. This concept has also been described in Australian patients, who believed to lack the authority needed to persuade family members to attend screening and thought risk information directly from healthcare would have “a bigger impact” [52]. 

### 4.3. Surprise about a Missing Link in the Triangle of Stakeholders

By adopting a pragmatic view on the duties and rights of stakeholders involved in risk disclosure, respondents identified a void, or a missing link, between two of them (Figure 1). The consequence of understanding the overwhelming duty of the index case created an expectation in the groups that “someone else” needed to step in and inform. We find this projection of duty to resemble previous findings reported as the *deferment of responsibility* [52], a way to *other* the responsibility [46] or having a *limited sense of duty* [47].

The idea that genetic risks should be handled more systematically by healthcare is not unique to our participants or context. Our research group has previously reported a strong public opinion in Sweden in favor of healthcare-mediated disclosure [29] but results from other counties support this as well. US respondents thought “direct contact should be a programmatic effort” [31], Australian patients thought that “information on screening should be disseminated through medical professionals” [52] and a recent Dutch systematic review on patient attitudes concluded that “actively offered support of healthcare professionals was desired” [36]. 

Although some informants considered the task of disclosure straightforward at first, when discussions progressed to more complicated family situations an assumption surfaced that healthcare would have a backup plan for reaching those who the index case had not informed. When presented with the current practice (family-mediated disclosure), expressions of surprise, concern for equal access to preventive care for all and questions on the appropriateness of leaving relatives at the will of the index case were voiced. These findings revealed a dissonance between the services participants assume would be available from healthcare and the actual services offered today by clinical genetics in Sweden (Figure 1). This “missing link” has been the topic of an ongoing debate for years, if healthcare should adopt a more proactive strategy for cascade testing [6,31,47,52]. Until now, most studies on attitudes towards direct-contact approaches have mainly focused on the perspective of patients or HCPs [53]. This could be a reason why the phenomenon of surprise about current practice, to our knowledge, has not been previously described. How this discrepancy affects the disclosure process and interaction with the genetic counsellor remains to be investigated.

### 4.4. More Personalized Information to Allow Informed Decisions

Our respondents clearly solicited easily accessible and concrete risk information with visual aids to make the message understandable to laypeople. Focus group studies from Canada and Australia have also reported wishes for clear, brief and graphically illustrated risk information [28,54]. Besides these wishes, our respondents voiced a clear desire for information to be highly personalized, preferably based on their family situation and to include contact information to easily accessible HCPs. Wishes were motivated by principle-based arguments such as being given the possibility of making an informed choice but also emotional arguments such as attaining “peace of mind” by understanding the details of the situation. 

Our findings align well with existing research on people’s motivations to know about hereditary risks. Three types of reasons for seeking out genetic risk information have previously been described: First, clinical utility and possibility of prevention in the family [19,20,21]. Second, gaining more information to reduce uncertainty and create a sense of control [21,22]. Third, feeling entitled to personal data and expecting to be informed about actionable findings from an authority or institution [22].

### 4.5. Strengths and Limitations

As qualitative researchers we acknowledge that data could have multiple meanings and that authors pre-understandings may influence the interpretation. The methods used in this study were chosen to describe the variance in perceptions of genetic cancer risk communication, not to quantify how common a certain view was.

Measures to ensure trustworthiness included member checks with verbal summaries during interviews, sessions with parallel coding and consensus discussions of emerging codes and categories by all authors [55]. To preserve context in the decontextualization phase, we used the full transcripts of interview data as unit of analysis and kept a record of analytical notes and memos throughout the process (14). To identify nuances and underlying meaning in the data, we explored latent content to document findings not initially evident in the manifest material [42,56,57].

Sampling data over time may have contributed to reducing risk of temporary influences and increased the dependability of our findings. Further measures to boost dependability included the use of early memos, field notes and multi-professional perspectives in the analysis [56]. Transferability may be limited as respondents were all Scandinavian. We conducted the discussions according to parameters known to positively affect discussion climate and small groups of 3–4 participants, meeting in a neutral office environment, held without breaks and offering ample time for each group [38]. Despite the inherent issue of sampling opinions through vignettes, our study design enabled us to catch naïve opinions from the public, as this would have been impossible in patients who have undergone genetic testing, since experience may affect attitudes [58].

## 5. Conclusions

Members of the public consider hereditary information a complex matter, with emotional and inter-personal consequences at stake. By limiting the individual obligations to well established relations and transferring some responsibility onto healthcare, expectations emerged about an institutional follow-up of at-risk relatives not currently used in clinical genetics in Sweden. This highlights a discrepancy between public expectations and clinical practice revealing a missing link in the communication between two of the three stakeholders involved in risk disclosure. Unmet expectations could be problematic in the short term as they can impede communication and in the long term as they can create mistrust for the institution of healthcare.

A better understanding of expectations between patients, healthcare and relatives, and a more collaborative approach towards risk disclosure, might be a way forward in improving outcomes in targeted cascade screening for hereditary cancer. Future research should investigate HCPs’ perspectives on how they see themselves assisting both index cases and relatives to fulfil the mission of disclosing risk information to those at risk.

Finally, there is a need to evaluate protocols for genetic disclosure which includes the individual patient in the decision-making of the risk disclosure process, co-creating the contact strategy for each family constellation and actively offer alternatives to the prevailing family-mediated disclosure pathway. Our findings pinpoint opportunities for clinical development which may prove crucial in reaching the full potential of personalized prevention in hereditary cancer and other tier-1 hereditary conditions.

## Figures and Tables

**Figure 1 jpm-11-01191-f001:**
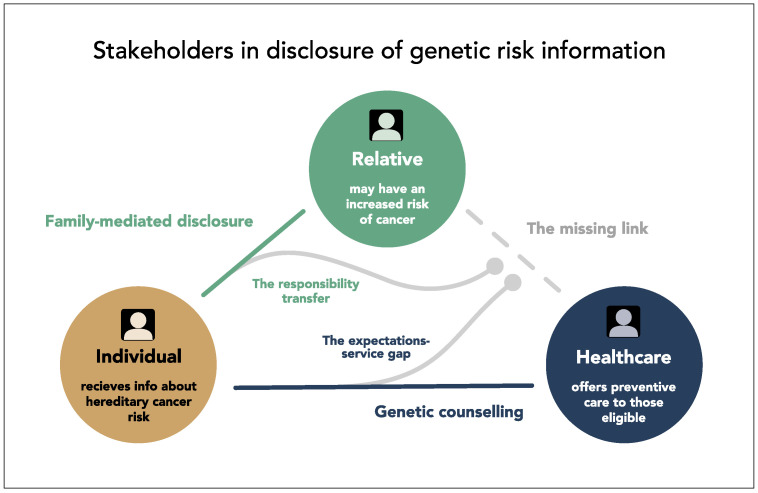
Conceptualization of stakeholders and their relationships based on our results.

**Table 1 jpm-11-01191-t001:** The initial steps in qualitative content analysis illustrated with two examples of identified meaning units.

Meaning Unit	Condensation	Abstraction	
Text	Condensed text	Code	Sub-category
…I think there’s a risk, like I said before, that one might feel a lot of guilt when talking about it, that one think’s it’s my fault passing on this or subjecting someone to…	might feel guilty when disclosing risk info think it’s my fault for passing it on	Feeling guilty	Uncomfortable to talk about cancer risk
…I would like to hear it from healthcare…those who work with this…I mean not that you’re supposed to go google this…	like to hear information from HCP not search information by oneself	Wanting information from healthcare	Prefer healthcare to handle disclosure

**Table 2 jpm-11-01191-t002:** Characteristics of study participants per focus group discussion (FGD).

	Characteristic	Total	FGD1	FGD2	FGD3	FGD4	FGD5
Gender	Men	8	1	1	1	3	2
	Women	10	3	3	3		1
Education	Elementary school or less (<9 years)	2		1		1	
	12 years of school completed	5		2	2	1	
	Graduate studies (>one year)	3	1		1	1	
	University degree/higher education diploma	8	3	1	1		3
Age	20–29 years	4		1	1		2
	30–39 years	2	1				1
	40–49 years	2		2			
	50–59 years	4	2		2		
	60 or older	6	1	1	1	3	

**Table 3 jpm-11-01191-t003:** Categories and themes derived from the qualitative content analysis of focus group data.

Categories	Descriptive Themes	Overall Theme
Struggle with unpleasant feelings and consequences Allow type of relationship to govern preferences Recognize that disclosure requires skill and experience	Face an important but difficult challenge	A maze of challenges in a haze of silent expectations
Depend on healthcare to take main responsibility Surprise and frustration about current practice Want personalized info to enable informed choice	Expect healthcare to lead but also support disclosure	

## Data Availability

The data presented in this study are available on request from the corresponding author. The data are not publicly available due to confidentiality requirements and the personal nature of the research topic and questions discussed.

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
