# Peer review of "A Focus Group Study of Perceptions of Genetic Risk Disclosure in Members of the Public in Sweden: “I’ll Phone the Five Closest Ones, but What Happens to the Other Ten?”"

_jpm, 2021, doi:10.3390/jpm11111191_

Round 1
Reviewer 1 Report
This is a well written manuscript investigating an area of active research in clinical genetics that will be an excellent addition to the literature. I would urge the authors to consider the following recommendations:
Major:
Consider discussing literature on public attitudes to cancer genetic testing (there are data from population surveys) as well as barriers and facilitators of family communication of genetic information. If length is a worry, the section on “right to know” in the introduction can be shortened as this is a study of potential index patients (not potential relatives who will be informed). If you did not intend to study index patients (which the discussion suggests), please state that explicitly in the introduction.
Consider adding an explanation of why data were collected over a two year time period. The extended time period is concerning as genetic testing is a dynamic field with well documented changes in public attitudes and awareness to testing over time.
Relatedly, please add a justification for why a “small group size of 3-4 participants per session” chosen for this study. And why 5 focus groups were deemed to be adequate? Sample size in qualitative work is often determined through saturation, is that what the authors used in this study?
Minor:
While accurate, did describing hereditary cancer risk as “generic risk of treatable cancer that may or may not be inherited by subsequent generations” cause confusion for participants?
Consider elaborating the short introduction on hereditary cancer risk that was provided to participants. Conceptualization of different cancers, e.g., cancers that affect mostly females such as breast may elicit different responses regarding family communication than Lynch Syndrome which affect males and females equally.
Line 79: pathogenic, not pathologic variant.
Author Response
Reply to Reviewer 1
Comment 1: "This is a well written manuscript investigating an area of active research in clinical genetics that will be an excellent addition to the literature. I would urge the authors to consider the following recommendations: "
Author reply - C1: Thank you for your time and effort to review our work. We are very thankful for each point of feedback that can improve our manuscript. We have considered all your comments in detail and made alterations and corrected the draft accordingly.
N.B. All alterations and additions in the manuscript have been highlighted in GREEN in the second version of our draft.
Comment 2: "Major: Consider discussing literature on public attitudes to cancer genetic testing (there are data from population surveys) as well as barriers and facilitators of family communication of genetic information."
Author reply - C2: Thank you for pointing this out. We have reviewed and re-written the relevant section in the introduction to more clearly outline which literature concerns attitudes on hereditary risk in general and what opinion studies discuss cancer genetic testing in particular in samples using members of the public. If there are particular works we have not identified but you find critical to refer to, please feel free to comment further.
Comment 3: "If length is a worry, the section on “right to know” in the introduction can be shortened as this is a study of potential index patients (not potential relatives who will be informed). If you did not intend to study index patients (which the discussion suggests), please state that explicitly in the introduction."
Author reply - C3: This feedback is very important and valuable to us, as this concept of shifting perspectives (between acting as a relative receiving information, and an index-case being the first one in the family to test positive for hereditary cancer) is one of the major findings in our data. We did, in fact attempt to primarily document attitudes of "potential relatives who will receive risk information" - but our respondents were so quick to switch to a "proband-role" (as in informing other family members) - that a lot of the data concerned both perspectives, which was also very valuable. However, to avoid confusion as to what our aim was, we have now ADDED a few clarifications to the last section of the Introduction stating which perspective was our primary focus to study.
Comment 4: "Consider adding an explanation of why data were collected over a two year time period. The extended time period is concerning as genetic testing is a dynamic field with well documented changes in public attitudes and awareness to testing over time."
Author reply- C4: A very good point indeed. The dynamic character of the field - and potentially changing public attitudes - is also why we did chose to add more focus group sessions after two years. As the content analysis progressed on the initial four transcripts, we wanted to corroborate our preliminary findings by adding a few more participants with younger ages and at a later time point. However, the resulting data was well in line with the earlier material and we could confirm we had reached a justifiable data saturation level. We have revised the methods section and described the data collection in more detail to clarify this point.
Comment 5: Relatedly, please add a justification for why a “small group size of 3-4 participants per session” chosen for this study.
Author reply - C5: Thank you for highlighting this methodological aspect. We have indeed considered the number of members in detail, as we were aware our topic might be considered personal and even private to talk about. The number of participants were thus planned to be between 3-5 in each group, but depending on availability and last minute changes we ended up having 3-4 participants consistently, which proved to be a good number for deeper elaborate discussions with room for everyone to have their turn to talk. We have added a sentence about this in the Methods section.
Comment 6: And why 5 focus groups were deemed to be adequate? Sample size in qualitative work is often determined through saturation, is that what the authors used in this study?
Author reply - C6: Indeed a good point to adress clearly. In one of our methodological references (Peek&Fothergill, 2009) the standard number of focus group is described as "Regarding the ‘ideal number’ of focus groups to conduct, most scholars agree that three to five groups are usually adequate, as more groups seldom provide new insights (Morgan, 1997; Krueger, 1988)." We chose to conduct five groups since we had quite a low number of participants in each. After the fifth group was conducted we could indeed confirm saturation of data, and dependability of some of the main findings over time, thus considered this total to be the appropriate number of groups for this particular research question. We have now clarified this in the methods section.
Comment 7: Minor: While accurate, did describing hereditary cancer risk as “genetic risk of treatable cancer that may or may not be inherited by subsequent generations” cause confusion for participants?
Author reply - C7: Thank's for this reflection. Indeed, risk communication is a difficult subject, and we did experience a few instances of initial misunderstandings in some groups. This was however resolved by the facilitators present, offering time to question and explain the vignettes before each participant could share their opinions on the matter. We have added a short description in the methods to better describe this interactive process.
Comment 8: Consider elaborating the short introduction on hereditary cancer risk that was provided to participants. Conceptualisation of different cancers, e.g., cancers that affect mostly females such as breast may elicit different responses regarding family communication than Lynch Syndrome which affect males and females equally.
Author reply - C8: Good point indeed. We have now elaborated a few sentences to more clearly describe the context and disease types under discussion. We've also added a reference to the Scenario Vignettes on the second page of Appendix 1 (where one can read the vignettes with using familial breast cancer risk as an example).
Comment 9: "Line 79: pathogenic, not pathologic variant."
Author reply - C9: Thank you for noting this spelling error. It has been corrected.
_________________________________________
Thank you again for the feedback and input!
Sincerely
C. Hawranek
Reviewer 2 Report
This is a well-conducted and clearly written report of a focus group study conducted with a convenience sample of members of the public about preferences for disclosure of hypothetical genetic test results.
I found the methods well presented, the analysis to be clearly written, and the literature review and discussion appropriate.
Author Response
Reply to Reviewer 2
Comment 1: "This is a well-conducted and clearly written report of a focus group study conducted with a convenience sample of members of the public about preferences for disclosure of hypothetical genetic test results.
I found the methods well presented, the analysis to be clearly written, and the literature review and discussion appropriate."
Author reply: Thank you very much for the positive feed-back. We are grateful for your time and the expertise offered to review our work.
/C. Hawranek